# Nutritional and Lifestyle Behaviors and Their Influence on Sleep Quality Among Spanish Adult Women

**DOI:** 10.3390/nu17132225

**Published:** 2025-07-04

**Authors:** Andrés Vicente Marín Ferrandis, Agnese Broccolo, Michela Piredda, Valentina Micheluzzi, Elena Sandri

**Affiliations:** 1Faculty of Medicine and Health Sciences, Catholic University of Valencia San Vicente Mártir, c/Quevedo, 2, 46001 Valencia, Spain; andres.marin@mail.ucv.es (A.V.M.F.); elena.sandri@ucv.es (E.S.); 2Department of Biomedicine and Prevention, Tor Vergata University, Via Montpellier, 00133 Rome, Italy; agnese.broccolo@alumni.uniroma2.eu (A.B.); valentina.micheluzzi@gmail.com (V.M.); 3Research Unit of Nursing Science, Department of Medicine and Surgery, Campus Bio-Medico di Roma University, Via Alvaro del Portillo, 21, 00128 Rome, Italy; 4Clinical and Interventional Cardiology, Sassari University Hospital, 07100 Sassari, Italy

**Keywords:** sleep duration, sleep quality, sleep hygiene, diet, food and nutrition, healthy lifestyle, socio-demographic factors, women, Spain

## Abstract

***Background:*** Sleep is a fundamental component of health, and deprivation has been linked to numerous adverse outcomes, including reduced academic and occupational performance, greater risk of accidents, and increased susceptibility to chronic diseases and premature mortality. Dietary and lifestyle behaviors are increasingly recognized as key determinants of sleep quality. Women are particularly susceptible to sleep disturbances due to hormonal fluctuations and psychosocial factors. However, women remain underrepresented in sleep research. This study aims to examine the associations between sleep quality, nutrition, and lifestyle in a large cohort of Spanish women. ***Methods:*** A cross-sectional study was conducted with 785 women aged 18–64. Participants completed the Pittsburgh Sleep Quality Index (PSQI) and the NutSo-HH questionnaire on dietary and lifestyle behaviors. Descriptive analyses, correlation matrices, Gaussian Graphical Models, and Principal Component Analyses were used to assess relationships between variables. ***Results:*** More than half of the participants rated their sleep quality as good or very good, although over 30% experienced frequent nighttime awakenings. Poor sleep quality was significantly associated with higher alcohol consumption, lower vegetable and white fish intake, and lower levels of physical activity. Diets rich in ultra-processed foods correlated moderately with subjective poor sleep and daytime dysfunction. However, no strong associations were found between stimulant consumption, late meals, or dietary patterns (e.g., Mediterranean diet) and sleep. Self-perceived health emerged as a protective factor, while nocturnal lifestyles were linked to longer sleep latency and fragmented sleep. ***Conclusions:*** In adult women, better sleep quality is linked to healthy dietary choices, regular physical activity, and a positive perception of general health. In contrast, alcohol use and irregular lifestyles are associated with poor sleep. Individual variability and cultural adaptation may moderate the impact of some traditionally harmful behaviors. Personalized, multidimensional interventions are recommended for promoting sleep health in women.

## 1. Introduction

Sleep is an essential physiological function, deeply involved in maintaining homeostatic balance and overall psychophysical health [1]. Adequate sleep duration and quality are associated with benefits across multiple systems, such as the cognitive, immune, metabolic, and cardiovascular systems, and are considered key determinants of both individual and population health [2]. Conversely, insufficient sleep duration and quality are linked to a wide range of negative outcomes, both health-related and social—such as poorer academic performance [3,4], reduced work productivity, and increased risk of workplace accidents [5]. Prolonged curtailment of sleep duration is a risk factor for the development of obesity, diabetes, hypertension, heart disease, and stroke and may contribute, in the long term, to premature death [6]. According to the revised Third Edition of the International Classification of Sleep Disorders, insufficient sleep syndrome is defined as a chronic pattern of reduced sleep duration, persisting for at least three months on most days of the week, and accompanied by daytime sleepiness that improves with increased total sleep time [7]. This condition occurs despite adequate opportunity and circumstances for sleep and is associated with significant daytime functional impairment [7]. Unlike insomnia, which is characterized by difficulty sleeping despite sufficient opportunity, insufficient sleep syndrome is generally the result of chronic, behaviorally induced sleep restriction, often driven by environmental, social, or occupational factors [8]. Due to its increasing impact, closely tied to lifestyle changes and shifts in societal rhythms, the World Health Organization and several scientific bodies have recognized this condition as an emerging public health priority [9,10].

Sleep quality and duration are influenced by a range of factors, including environmental (light and noise pollution, climate), social (poverty, insecurity, inequality), and behavioral (excessive use of electronic devices, shift work, stress) factors, as well as dietary habits, lifestyle, interpersonal factors (such as age and gender), and medical conditions such as chronic diseases and hormonal imbalances [9,11,12]. Women appear to be particularly vulnerable to sleep disturbances, with gender differences in sleep emerging early in life [13]. Compared to men, women consistently report poorer sleep quality and a higher risk of developing insomnia [14]. These disparities are influenced by multiple factors, including hormonal fluctuations, psychological stress, depression, aging, and shifting life roles [14]. During the menstrual cycle, especially in the premenstrual phase, alterations in circadian rhythms and sleep quality are commonly observed, even in the absence of clinically significant menstrual symptoms. Women with severe premenstrual syndrome often experience disturbing dreams, fatigue, and reduced concentration [15]. Sleep disturbances are also highly prevalent during pregnancy and tend to increase as gestation progresses. In the postpartum period, hormonal withdrawal combined with the newborn’s irregular sleep–wake cycles further contributes to maternal sleep fragmentation. Additionally, insomnia is one of the most frequently reported complaints among perimenopausal women, whose vulnerability to sleep disorders increases significantly during this transitional phase [16]. Recent research strongly highlights the connection between sleep and the phases of the female reproductive cycle, underscoring the importance of developing tailored therapeutic strategies for managing insomnia in women [17].

At the same time, there is a growing consensus in the scientific literature regarding the key role of diet and lifestyle in sleep regulation. Diets high in ultra-processed foods, simple sugars, and saturated fats are frequently associated with poorer sleep quality [18]. In contrast, dietary patterns rich in vegetables, whole grains, and omega-3 fatty acids are linked to deeper and more restorative sleep [19,20]. However, significant gaps remain in the literature, particularly concerning adult women, who are often underrepresented in sleep-related research. Given their distinct physiological, hormonal, and social characteristics, women may exhibit a particular vulnerability to sleep dysregulation, warranting targeted research.

In light of these considerations, the present study aims to analyze the associations between sleep quality and nutrition and lifestyle behaviors in a large sample of adult women residing in Spain. By employing validated instruments such as the Pittsburgh Sleep Quality Index (PSQI) [21] and the NutSo-HH scale [22], this study aims to provide a multidimensional characterization of the behavioral factors influencing sleep quality. More specifically, it aims to identify patterns associated with suboptimal sleep quality, thereby contributing to the development of evidence-based strategies for prevention and health promotion tailored to adult women.

## 2. Materials and Methods

### 2.1. Type of Study and Inclusion Criteria

A cross-sectional study was conducted on the Spanish women population residing in Spain. Women aged between 18 and 64 years were included in the sample. Individuals with chronic or acute medical conditions known to affect sleep quality were excluded, as well as those experiencing temporary situations that could interfere with their usual routines, such as hospitalization or incarceration. However, healthy individuals who might occasionally take some form of medication (natural or chemical) to help fall asleep were not excluded from the study.

The study focused on adult women because, on the one hand, women tend to show greater willingness and reliability when responding to health- and nutrition-related questionnaires. Previous research has consistently shown that women are more engaged in health-related studies and more likely to report their behaviors and symptoms accurately, which contributes to more robust and interpretable data [23,24,25]. On the other hand, scientific evidence from the Spanish adult population indicates that women experience significantly higher rates of sleep disorders, such as insomnia, and poorer sleep quality compared to men [26,27]. Therefore, targeting this population makes the study particularly relevant.

### 2.2. Sample Size Calculation

To estimate the required sample size, reference was made to the Continuous Population Statistics (ECP) provided by the Spanish National Institute of Statistics (INE) [28]. As of 1 January 2024, the total population in Spain was 21,544,520. Based on this figure and using the standard formula for sample size estimation in finite populations—with a 95% confidence level, 5% margin of error, and the assumption of maximum variability (*p* = q = 0.5)—a minimum of 385 participants was calculated to be necessary.

Finally, the total number of valid questionnaires collected was 785.

### 2.3. Ethical Considerations

The study was conducted in accordance with the ethical principles outlined in the Declaration of Helsinki [29] and received approval from the Research Ethics Committee of the Catholic University of Valencia (approval code UCV/2024-2025/060, dated 28 January 2025). Prior to completing the questionnaire, all participants were informed about the objectives of the study. They were assured that their responses would remain anonymous and that all data would be analyzed in aggregate form. Informed consent was obtained from each participant, including their agreement to the publication of the study’s findings.

### 2.4. Instruments

Two validated questionnaires were used to collect data. Qualitative questions were added to collect socio-demographic and other data of interest.

To assess sleep quality, the Spanish version [30] of the Pittsburgh Sleep Quality Index (PSQI) was used [21]. The PSQI is a widely used and validated instrument assessing sleep habits. Respondents are asked to report their sleep behavior (such as the time of going to bed or waking up, the time it took to fall asleep, or the number of times they got up during the night, among others) during most days and nights of the previous month.

The Spanish version of the PSQI consists of 19 questions grouped into the following seven items:Item 1: subjective sleep quality;Item 2: sleep latency;Item 3: sleep duration;Item 4: habitual sleep efficiency;Item 5: sleep disturbances;Item 6: use of sleep medication;Item 7: daytime dysfunction.

Each item is scored on a scale from 0 to 3 points, where a score of “0” indicates no difficulty, while a score of “3” indicates severe difficulty within that component. The total PSQI score is the sum of the seven component scores, ranging from 0 to 21. A total score of “0” indicates good overall sleep quality, while “21” reflects severe difficulties across all components.

The NutSo-HH (Nutritional and Social Healthy Habits scale) [22] gathers detailed information on participants’ nutritional, social, and lifestyle habits, focusing on their choices and frequency of consumption. This multidimensional instrument was developed through a rigorous validation process, including psychometric testing, to ensure its reliability and accuracy in capturing health-related behaviors. The scale is structured around six core factors: F1 (Mediterranean foods), F2 (healthy and unhealthy foods), F3 (meats and dairy products), F4 (eating disorders), F5 (rest habits), and F6 (alcohol consumption). Additionally, two broader categories are derived: NUTRI (combining F2 and F3) and HH (combining F4 and F5). To reduce the risk of data loss due to user error, most questions were designed as closed-ended, multiple-choice items, enabling respondents to select the option that best described their habits and consumption patterns.

The survey also included questions on socio-demographic characteristics, as well as a set of qualitative items. Two questions focused on physical activity, assessing the frequency and duration of exercise sessions. Additionally, three items explored eating behaviors commonly thought to impact sleep quality, such as having meals close to bedtime, consuming heavy evening meals, and using stimulants during the day (e.g., tea, coffee, energy drinks, caffeinated beverages, or supplements).

### 2.5. Data Collection

Potential participants were recruited via email and social media, with a particular focus on the Instagram account @elretonutricional. This platform was used to engage health professionals and influencers who, after being informed about the project, shared the questionnaire through their own communication channels. The research team also used their personal and professional networks on LinkedIn, Twitter, WhatsApp, and Facebook to broaden the survey’s reach. Additionally, students from the Catholic University of Valencia—many of whom come from various regions across Spain—actively collaborated in sharing the questionnaire within their own circles, significantly contributing to national-level outreach. To enhance participation, a snowball sampling strategy was employed [31], whereby respondents who met the inclusion criteria were encouraged to forward the survey to others with similar profiles. This chain-referral process was repeated until the desired sample size was achieved. The study followed the STROBE guidelines for observational research [32], and data collection was conducted between February and May 2025.

### 2.6. Variables

#### 2.6.1. Socio-Demographic Variables

The questionnaire collected detailed information on a range of socio-demographic variables, including age, place of residence, educational level and field of study, income level, and living situation (i.e., living alone, with others, or in the family home).

Age was categorized into four groups based on the classification by Medley et al. [33]: young (18–22 years), early adulthood (23–34 years), early middle age (35–44 years), and late middle age (45–64 years). Educational attainment was grouped into two categories: basic education (no formal education, primary or secondary education, vocational training, or high-school diploma) and higher education (bachelor’s, master’s, or doctoral degree). Participants were also asked whether they had completed or were currently pursuing higher education in a health-related field.

Income was classified into three categories: low income (≤EUR 2200/month), medium–high income (>EUR 2200/month), and no response. Living arrangements were categorized as either living alone or with others, while family living was further specified as living in the family home versus living independently.

#### 2.6.2. Nutritional and Lifestyle Variables

Following the criteria used in previous articles [34,35,36], nutrition and lifestyle variables were categorized as shown in Table 1.

#### 2.6.3. Sleep Quality Variables

Following the Pittsburgh Sleep Quality Inventory (PSQI) test scoring instructions, the sleep quality variables were categorized as shown in Table 2.

The global PSQI score was categorized into three ranges to facilitate interpretation: scores from 0 to 7 were classified as indicating good sleep quality, scores from 8 to 14 reflected sleep quality in need of improvement, and scores from 15 to 21 were considered indicative of poor sleep quality.

### 2.7. Data Analysis

The dataset was initially preprocessed by removing invalid, inconsistent, or extreme entries, including inaccurate responses and statistical outliers. Following this data cleaning process, the Shapiro–Wilk test was employed to assess normality, revealing that none of the variables followed a normal distribution. These findings were further confirmed through the visual inspection of Q–Q plots [37]. Due to this lack of normality, appropriate non-parametric statistical methods were applied: the Chi-square test was used for categorical variables, and the Kruskal–Wallis test was employed for ordinal or continuous variables across independent groups. Categorical socio-demographic variables are presented as frequencies and percentages, while continuous variables are summarized using means and standard deviations. A *p*-value of less than 0.05 was considered statistically significant in all analyses.

To further investigate the associations between dietary behaviors and sleep quality dimensions, a network analysis was performed. In the resulting network diagram, variables are represented as nodes (circles), and relationships between them are depicted by edges (lines), whose thickness and color indicate the strength and direction of the correlations.

Spearman’s correlation coefficients were used to define edge weights. To isolate direct relationships and reduce the influence of confounding variables, we implemented a Gaussian Graphical Model (GGM) with model selection based on the Extended Bayesian Information Criterion (EBIC) [38]. This technique estimates partial correlations, allowing for a clearer understanding of the unique associations between variables.

The EBIC regularization enhances the model by improving sparsity and minimizing overfitting, thereby highlighting the most meaningful and robust associations. This method facilitates the interpretation of complex variable interactions, even for non-specialist audiences.

Principal Component Analysis (PCA), an unsupervised learning technique, was employed to reduce the dimensionality of the dataset by summarizing multiple variables into a smaller number of components that retain most of the original information. The resulting PCA plot helps uncover patterns, groupings, and variable relationships. In this visual representation, arrows indicate the contribution of each variable: longer arrows reflect greater influence on a component, while their direction reveals correlation patterns—variables pointing in the same direction are positively associated, and those in opposite directions are negatively related. Perpendicular vectors suggest no correlation or limited relevance to that axis.

All statistical analyses were conducted using Jamovi (Version 2.4.44). The Gaussian network and the Principal Component Analysis graphics were generated within Jamovi, while supplementary graphical representations were produced using Microsoft Excel.

## 3. Results

### 3.1. Sample Composition

The initial dataset was cleaned according to the criteria indicated in Figure 1. Surveys with invalid data and those that did not meet the established inclusion criteria were eliminated. The final valid sample included 785 respondents.

### 3.2. Sample Description

The sample included women aged 18 to 64, distributed across four age groups: 19.0% were aged 18–22, 27.4% were in early adulthood (22–34), 26.2% were in early middle age (35–44), and another 27.4% were in late middle age (45–64). Most participants lived with others (90.2%) and with family members (80.6%). In terms of income, 32.5% reported low income, while 55.2% reported medium–high income. Higher education was completed by 61.9%; 35.5% had a background in health-related studies (see Table 3).

### 3.3. Analysis of the Sleep Variables of the Sample

Table 4 and Figure 2a–c offer a comprehensive overview of the participants’ sleep routines and contribute to the evaluation of their overall sleep quality. The most frequently reported bedtime was between 11:00 p.m. and midnight (42.7%), while the most common wake-up time was between 6:00 and 7:00 a.m. (39.7%), closely followed by the 7:00 to 8:00 a.m. slot (37.5%). Regarding sleep onset, 53% of respondents fell asleep in under 15 min, and 32.1% required between 15 and 30 min. More than half of the participants (54.8%) rated their sleep quality as good, with an additional 6.6% describing it as very good. Accordingly, 54.5% reported no issues with daytime functioning, and 33.4% experienced only occasional impairments in daily activities.

Figure 2a displays the monthly frequency of sleep-related difficulties experienced by adult women within the sample. Among the reported issues, nighttime or early morning awakenings were the most prevalent, with just 19.4% of participants reporting no such occurrences and 34.8% experiencing them three or more times per week. Trouble initiating sleep within the first 30 min was also notable, with 15.4% affected multiple times weekly. Additionally, 23.1% reported waking up to use the toilet at least three times per week. Other problems—such as respiratory difficulties, snoring, feeling too hot or cold, nightmares, and pain—were less frequently reported. The use of sleep medication was low, with 85% of respondents indicating no usage in the past month. Similarly, excessive daytime sleepiness during tasks like driving or eating was rare, reported as absent by 79% of participants. These findings indicate that while some sleep disruptions are relatively common, severe disturbances and impairments in daily functioning are generally infrequent in this group.

Table 4 and Figure 2b summarize the average scores for each of the seven components of the Pittsburgh Sleep Quality Index (PSQI) as reported by adult women in Spain. Among these, the highest mean score was recorded for subjective sleep quality (1.67), pointing to a moderate level of dissatisfaction with perceived restfulness. Sleep disturbances (1.23), duration (1.13), and latency (1.08) also showed high values, indicating difficulties with maintaining uninterrupted sleep and initiating it in a timely manner. In contrast, components such as daytime dysfunction (1.04) and habitual sleep efficiency (0.97) reflected comparatively more favorable sleep patterns, suggesting a lesser impact on daytime performance. The lowest score was observed in the use of sleep medication (0.42), showing that pharmacological intervention was infrequent. Taken together, these results indicate that while the reliance on sleep aids is low, perceived sleep quality and continuity remain key areas of concern in this population.

Figure 2c illustrates the distribution of total scores on the Pittsburgh Sleep Quality Index (PSQI) among the study population. Notably, no participants achieved a score of 0 or 1, indicating that none reported optimal sleep quality. Most scores were concentrated in the lower third of the scale, suggesting that most respondents experienced relatively good sleep. The mean global PSQI score was 7.5 (out of a maximum of 21), with a standard deviation (SD) of 3.5. Only a small proportion (<4%) reported very poor sleep quality, defined as a PSQI score >14.

### 3.4. Relationship Between Nutrition and Lifestyle Variables and Sleep Quality Variables

Table 5 compares the mean scores of dietary habits, health indicators, and lifestyle behaviors across three groups classified by sleep quality: good, needing improvement, and poor, based on PSQI scores. Statistically significant differences (*p* < 0.05) were observed in several variables.

Participants with good sleep quality reported significantly higher consumption of vegetables (M = 3.57, SD = 0.67) compared to those with poor sleep (M = 3.05, SD = 0.91; *p* = 0.004). Similarly, white fish intake was higher among those with good sleep (M = 1.75, SD = 0.56) than in the poor sleep group (M = 1.42, SD = 0.51; *p* = 0.041). While fruit and legume consumption were also slightly lower in the poor sleep group, these differences did not reach statistical significance.

Self-reported health perception showed a strong association with sleep quality, with the good sleep group reporting the highest score (M = 3.95, SD = 0.71) and the poor sleep group the lowest (M = 3.05, SD = 0.78; *p* < 0.001). Likewise, physical activity (PA) level was significantly higher among those with good sleep (M = 1.81, SD = 0.89) and lower in the poor sleep group (M = 1.26, SD = 0.93; *p* < 0.001).

Interestingly, although the average training time was longest in the good sleep group (M = 54.6 min), followed by the poor sleep group (M = 52.6 min), this difference was still statistically significant (*p* < 0.001), possibly reflecting variance in training consistency rather than duration alone.

Regarding risk behaviors, getting drunk showed a significant relationship with sleep quality and was more frequent among those with poor sleep (M = 1.11, SD = 0.32; *p* < 0.001). Other variables such as soft drink, fast food, or ultra-processed food intake showed a tendency toward higher consumption in the poor sleep group but did not reach statistical significance.

These findings suggest that healthier eating patterns, better physical health, and higher levels of physical activity are positively associated with better sleep quality, while frequent binge drinking appears to be a marker of poor sleep.

Figure 3 presents the correlation matrix between food consumption patterns and sleep quality indicators, as measured by the PSQI and its seven components. In general, most food categories exhibited weak correlations with sleep variables. However, a few noteworthy associations were observed, particularly with the intake of ultra-processed foods, fried food, and fast food.

Ultra-processed food consumption was positively correlated with subjective sleep quality (r = 0.429), daytime dysfunction (r = 0.454), and the global PSQI score (r = 0.503), suggesting that higher intake of these foods is associated with poorer sleep quality and increased daytime impairments. Similar patterns were found for fried food, which also showed positive correlations with subjective sleep quality (r = 0.430) and daytime dysfunction (r = 0.416). Fast food consumption was moderately associated with daytime dysfunction (r = 0.299), indicating a possible impact on alertness or functional capacity.

In contrast, fruit and vegetable intake displayed weak and non-significant negative correlations with most sleep parameters, though their relationships were minimal. Interestingly, soft drink consumption showed a negative correlation with sleep duration (r = −0.113), though this association was not statistically significant. Additionally, there was no clear relationship between self-reported dietary patterns (DIET variable) and any component of sleep quality.

Taken together, the results suggest that diets high in ultra-processed and fried foods may be linked to poorer perceived sleep and greater daytime dysfunction, whereas healthier dietary patterns (e.g., fruit and vegetable intake) did not show a strong association with sleep in this sample.

A Gaussian network analysis was conducted to evaluate the relationship between diet type (Mediterranean, plant-based, intermittent fasting, or others) and sleep quality (Figure 4). The results showed no significant association between dietary patterns and sleep components. Even eating habits typically considered influential—such as late dinners or consumption of stimulants—did not demonstrate a relevant impact. These findings suggest that, within this sample, diet and certain eating behaviors do not noticeably affect sleep quality.

Figure 5 displays the correlation matrix between sleep quality indicators and selected health and lifestyle behaviors, including physical activity, substance use, and self-perceived health. Overall, the data reveal several weak-to-moderate associations, particularly involving alcohol consumption, sleep latency, and subjective sleep quality.

Self-perceived health showed weak negative correlations with several components of poor sleep, including sleep latency (r = −0.240), daytime dysfunction (r = −0.231), and the global PSQI score (r = −0.280), suggesting that individuals with better perceived health tend to report better sleep outcomes.

Physical activity (PA) level and average training time per session were positively correlated with one another (r = 0.595), but neither showed strong relationships with sleep components. However, PA level had a weak negative correlation with the PSQI total score (r = −0.150), indicating that more active individuals may experience slightly better sleep quality.

Going out at night and alcohol consumption were both positively correlated with one another (r = 0.289) and with getting drunk (r = 0.239 and r = 0.226, respectively). These variables, particularly getting drunk, showed weak positive associations with sleep latency (r = 0.239), suggesting potential disruptions in sleep initiation related to alcohol use.

Importantly, subjective sleep quality was negatively correlated with sleep latency (r = −0.420), sleep duration (r = −0.490), sleep efficiency (r = −0.412), and daytime dysfunction (r = −0.336), indicating that poorer perceived sleep was closely tied to these objective and functional sleep components.

The global PSQI score correlated most strongly with sleep duration (r = 0.748), sleep efficiency (r = 0.727), and sleep latency (r = 0.562), confirming that these elements are the most influential factors driving overall sleep quality in this sample.

Taken together, these results suggest that while health-related behaviors such as physical activity and alcohol use have some influence, perceived sleep quality and its associated disturbances are more directly shaped by the physiological components of sleep.

### 3.5. Principal Component Analysis (PCA) Between Nutritional and Sleep Quality Variables

The PCA biplot in Figure 6 provides a visual representation of the relationships between dietary habits and sleep quality indicators in adult Spanish women (Dim1: 27%. The first two dimensions explain a combined variance of approximately 42.9.0%, Dim2: 15.9%) (The percentage of variance explained by each component is reported in Table A1, Appendix A).

Dimension 1 (Dim1) appears to distinguish between healthy dietary patterns and poorer sleep outcomes. Variables such as vegetable and fruit consumption, along with white fish intake, load negatively on this axis, while higher scores on sleep disturbances (Item 5), use of sleep medication (Item 6), daytime dysfunction (Item 7), and consumption of fast food, fried food, and ultra-processed food load positively. This indicates a potential inverse relationship between healthy eating behaviors and poor sleep outcomes.

Dimension 2 (Dim2) seems to capture a general sleep quality gradient. Positive loadings are observed for sleep duration (Item 3) and sleep efficiency (Item 4), which are classic indicators of good sleep. Conversely, subjective sleep quality (Item 1) loads negatively, suggesting it may reflect perceived sleep dissatisfaction despite objective parameters.

The direction and proximity of vectors imply meaningful associations:

Fruit, vegetables, and white fish cluster together, indicating they are commonly consumed in tandem.

Fast food, fried food, and ultra-processed food form a distinct group, aligned with sleep disturbances and dysfunction, hinting at a shared behavioral profile.

The near orthogonality between the healthy food cluster and poor sleep indicators suggests that these behaviors may operate independently in some individuals while being inversely related in others.

In summary, this PCA reveals two major behavioral patterns: one characterized by healthier dietary choices and better perceived sleep and another marked by high consumption of processed foods and sleep-related difficulties. These insights highlight potential intervention targets for promoting both sleep quality and nutritional health in women.

### 3.6. Principal Component Analysis (PCA) Between Lifestyle Variables and Sleep Quality Variables

The Principal Component Analysis (PCA) shown in Figure 7 explored the associations between lifestyle behaviors and sleep quality indicators (the percentage of variance explained by each component is reported in Table A2, Appendix A). The first two principal components explain 35.9% of the total variance (Dim1: 23.1%; Dim2: 12.8%), allowing for a simplified yet informative view of underlying patterns.

In the biplot, vectors pointing in similar directions represent positively correlated variables. Notably, Item 3 (sleep duration) and Item 4 (sleep efficiency) are closely aligned, suggesting they often co-occur. These are inversely related to behaviors such as smoking, alcohol consumption, and going out at night, which cluster together in the upper quadrant, indicating a negative association with good sleep outcomes.

Item 1 (subjective sleep quality) and PA level (physical activity) are aligned and opposed to sleep disturbances and use of medication, implying that more physically active individuals tend to report better perceived sleep quality and lower reliance on sleep aids.

The grouping of alcohol, going out, and getting drunk reveals a lifestyle component that may contribute to poorer sleep indicators such as latency, disturbances, and daytime dysfunction. This PCA highlights two main behavioral clusters—health-promoting habits versus risk-associated social behaviors—and their respective associations with sleep quality.

## 4. Discussion

This study analyzed dietary habits and lifestyle in relation to sleep quality in a sample of adult women residing in Spain, highlighting how specific behaviors may influence nighttime rest. The sample was heterogeneous in terms of age, living conditions, education level, and socioeconomic status. In particular, the prevalence of women with higher education (61.9%) and training in the health sector (35.5%) suggests a relatively high level of awareness regarding the importance of a healthy lifestyle. This finding is consistent with the literature, where socio-demographic factors such as education and income are associated with both sleep quality and the adoption of healthy behaviors [39].

Despite the generally positive subjective perception of sleep, some relevant issues, such as frequent nocturnal or early morning awakenings, were reported by over 30% of the participants several times per week. These disturbances are in line with the literature, which identifies nocturnal awakenings as one of the most common sleep issues in adult women, frequently associated with factors such as stress, hormonal changes, and environmental conditions [17,40,41,42]. Sleep fragmentation, regardless of the presence of daytime sleepiness, has been widely correlated with negative cognitive and psychophysical outcomes, including memory reduction, cognitive decline, persistent fatigue, and impaired quality of life [43,44,45,46]. Another factor that emerged frequently is nocturia, which is often underestimated but documented in the literature as one of the main causes of sleep interruption in adult women, with potential long-term consequences on well-being [47]. Therefore, even in the presence of a generally positive subjective perception, there are recurring signs of nocturnal disturbance—particularly awakenings and nocturia—that could have a cumulative impact on physical and cognitive health. This discrepancy between perceived and functional sleep raises the question of whether subjective assessment alone is sufficient to accurately describe sleep quality [48].

Although 61.4% of participants rated their sleep as “good” or “very good,” the mean score for subjective sleep quality on the Pittsburgh Sleep Quality Index (PSQI) was the highest among the seven components (M = 1.67), indicating a moderate level of dissatisfaction. This misalignment suggests a possible normalization of mild sleep disturbances, such as frequent awakenings or difficulties falling asleep, especially when these do not have evident repercussions on daytime functioning [49]. In other words, many participants seem to consider their sleep “good enough” if it does not interfere with their daily activities. This attitude has also been observed in previous studies, which indicate that adaptive or cultural factors may influence the subjective evaluation of sleep, particularly among adult women who are often exposed to chronic stress, family responsibilities, or age-related physiological changes [50,51,52]. These findings reinforce the importance of using standardized tools such as the PSQI to detect nuances in sleep quality that may not emerge through general self-reported measures. Although subjective sleep perception is sometimes more negative than objective measurements, it nonetheless remains a clinically relevant predictor of mental well-being and quality of life [53].

The comparative analysis between groups categorized according to sleep quality revealed a significant association between a diet rich in vegetables and fish and better sleep quality, as confirmed in the literature [54,55]. Various foods provide essential micronutrients that can positively influence the neurobiological processes involved in sleep regulation. B vitamins, such as B6, B9 (folate), and B12, play a role in the synthesis of key neurotransmitters for sleep, including serotonin and melatonin [55]. Omega-3 fatty acids, particularly DHA, have been shown to improve sleep quality and modulate circadian processes [56]. Furthermore, high intake of antioxidants, such as vitamins C and E, has been associated with a reduced risk of sleep disturbances, suggesting a neuroprotective role in the regulation of sleep [57].

It is also plausible that a higher-quality diet reflects a broader set of healthy behaviors; for example, women who follow a balanced diet may have a more regular lifestyle, lower levels of perceived stress, and greater attention to overall health—all factors known to positively influence sleep quality [19].

The perception of good general health and higher levels of physical activity are significantly associated with better sleep quality, in line with previous studies [58,59,60]. Regular physical activity promotes sleep through various physiological mechanisms: it supports more efficient regulation of body temperature, reduces cortisol levels, and improves heart rate variability—all factors that facilitate the initiation and maintenance of sleep [61]. Specifically, physical exercise has been associated with increased duration of deep sleep (NREM phase) and reduced nocturnal awakenings, thereby improving sleep efficiency and perceived sleep quality [40,62].

Good-quality sleep has been found to contribute to a better perception of health status, in terms of energy levels, mood, and cognitive functioning during the day [63]. Conversely, high levels of perceived stress are correlated with poor sleep quality and the onset of insomnia, with anxiety and depression partially mediating this relationship [64]. Therefore, the perception of good health tends to be associated with lower levels of stress and anxiety, which are well-known factors that negatively affect sleep quality [65]. This relationship suggests the existence of a virtuous cycle in which a better perception of overall health is associated with greater psychological stability and higher sleep quality, reducing the perceived need to resort to sleep medications.

A negative correlation emerged between perceived health and several parameters indicative of sleep disturbance, including sleep onset latency, daytime dysfunction, and the global score on the Pittsburgh Sleep Quality Index (PSQI). Women who reported a subjectively good health status also tended to experience more efficient sleep, with fewer difficulties falling asleep and reduced daily impact from sleep disturbances. These results align with a growing body of studies that highlight a bidirectional relationship between sleep quality and general psychophysical health: good-quality sleep promotes physical and mental well-being, and in turn, a perceived positive health status favors more regular and restorative sleep [66,67]. It is plausible that a positive evaluation of one’s own health reflects, at least in part, greater psychological resilience, the presence of healthy behavioral habits, and lower exposure to stress factors—all elements that facilitate the reduction in sleep latency and improve the subjective perception of sleep [68,69].

Alcohol abuse emerges as a factor significantly associated with compromised sleep quality, in line with studies documenting that alcohol negatively affects sleep latency, continuity, and architecture [70]. Even moderate consumption can increase sleep fragmentation, reduce REM sleep, and cause daytime sleepiness [71]. Although alcohol may initially facilitate falling asleep due to its sedative effects, it subsequently alters sleep structure, leading to lower-quality sleep [72]. This can trigger a vicious cycle in which alcohol is used as a strategy to aid falling asleep—especially among those with chronic insomnia—but in the long term, it contributes to worsening sleep quality [73]. This self-medicating use of alcohol is considered a maladaptive strategy, associated with an increased risk of persistent sleep disorders and dependence [74].

The positive correlation between alcohol consumption, frequency of evening outings, and increased sleep latency further suggests that behaviors related to a nocturnal lifestyle can negatively interfere with the neurophysiological mechanisms of sleep. Alcohol, while it may temporarily reduce sleep latency, compromises the integrity of sleep architecture in the following hours [75], while frequent nighttime outings can expose individuals to sensory stimulation, artificial light, and irregular circadian rhythms—all factors known to disrupt sleep [76]. Therefore, addressing this type of behavior represents a clinical priority in the promotion of good sleep hygiene.

Although some less healthy dietary behaviors appeared to be more frequent among participants with lower sleep quality, these associations were not statistically significant. It is possible that the limited sample size or high individual variability may have attenuated the emergence of real relationships, as observed in other studies where associations between diet and sleep appear complex and are mediated by numerous factors [77].

Regarding physical activity, a strong correlation was observed between weekly frequency and average session duration (r = 0.595), but both showed only weak correlations with perceived sleep quality. These results suggest that it is not the total amount of exercise that directly influences sleep quality but rather qualitative characteristics such as the type of activity, its intensity, and the time of day in which it is performed [59]. Physical exercise, regardless of the time it is carried out, is associated with a prolongation of NREM sleep and an overall improvement in sleep quality [59]. However, although light-to-moderate-intensity physical activity is particularly effective in promoting restorative sleep, it is still advisable to avoid vigorous exercise in the late evening hours, as it may interfere with falling asleep in sensitive individuals [62,78,79].

The Gaussian network analysis conducted on the sample did not show a significant association between the type of diet followed (including Mediterranean, vegetarian, intermittent fasting, or others) and the main parameters of sleep quality. This result suggests that, in this population, it is not so much the overall dietary pattern that influences sleep but rather the quality of specific foods and other lifestyle factors, such as physical activity, perceived stress, and the regularity of daily habits. The literature on this topic shows contrasting results. On the one hand, several studies attribute a protective effect to the Mediterranean diet against sleep disturbances due to its high content of anti-inflammatory nutrients and circadian rhythm regulators, such as omega-3s, fiber, and polyphenols [80]. On the other hand, a more complex relationship emerges, mediated by interacting factors such as individual metabolism, inflammatory status, and psychological well-being [81].

Unexpectedly, even behaviors commonly considered harmful to sleep—such as consuming large meals in the evening or the intake of stimulants like coffee, tea, or energy drinks—did not show statistically significant correlations with sleep quality in our sample. One possible explanation lies in marked interindividual differences in sensitivity to these factors, particularly to caffeine, whose metabolism is strongly influenced by genetic variants in the CYP1A2 gene. Individuals with fast caffeine metabolism may tolerate its intake even in the evening hours without evident repercussions on sleep [82]. Similarly, the timing of meals can have variable effects depending on food composition, synchronization with the individual circadian rhythm, and personal chronotype [83]. Therefore, it is plausible that some dietary behaviors were perceived as neutral or well tolerated by participants while potentially problematic in other contexts or clinical groups.

These results do not contradict established evidence on the negative effects of caffeine and evening meals on sleep quality [84,85] but rather highlight the importance of considering moderating factors and individual adaptations when evaluating dietary habits and their effects on sleep, particularly in adult and culturally specific populations.

### 4.1. Strengths and Limitations of the Study

This study presents several strengths. First, the large sample size (n = 785) and its heterogeneity in terms of age, education level, occupation, and lifestyle provide a comprehensive overview of sleep-related habits among the adult female population in Spain. Second, the use of validated and reliable instruments—such as the Pittsburgh Sleep Quality Index (PSQI) for assessing sleep quality and the NutSo-HH for analyzing nutritional and social behaviors—ensured systematic and robust data collection. Furthermore, the multidimensional approach, which integrated traditional statistical analyses with graphical models (GGM) and dimensionality reduction techniques (PCA), allowed for an in-depth exploration of the interrelationships between behavioral, socio-demographic, and health variables.

However, some limitations should also be acknowledged. The cross-sectional nature of the study does not allow for causal inferences between the variables examined. Additionally, given the exploratory and descriptive nature of the study, multivariate adjustments were not performed. This may limit the ability to control confounding effects and should be considered when interpreting the results. In addition, data were collected through self-reported questionnaires, which may be subject to social desirability and recall bias, particularly regarding sleep duration, dietary intake, and physical activity. Another limitation lies in the absence of objective sleep measurements, such as actigraphy or polysomnography, which could have validated the subjective sleep quality assessments.

Finally, the study did not include hormonal and psychological variables, which are known to influence sleep in women and could have provided additional insight into the mechanisms underlying sleep disturbances in this population. Future studies should consider integrating these dimensions into their analysis.

### 4.2. Public Health Implications of the Study

The results of this study open several perspectives for practical application in clinical and health promotion settings and for future research.

From an applied perspective, the findings support the importance of multidisciplinary and personalized interventions, including non-pharmacological approaches, that consider not only eating habits but also psychological aspects, stress levels, and physical activity patterns [60,86,87]. Educational programs aimed at promoting a balanced diet rich in beneficial micronutrients, regular physical activity, and proper sleep hygiene could represent effective strategies to enhance overall well-being and prevent sleep disorders in the adult female population. Attention to alcohol use and disorganized nighttime behaviors should be strengthened, given their association with compromised sleep quality. An additional promising development concerns the assessment of genetic and biological components that modulate individual responses to stimulants such as caffeine or to specific dietary patterns, with the aim of developing more targeted and personalized nutritional and behavioral recommendations. Furthermore, the use of digital technologies for continuous monitoring of sleep, diet, and physical activity could facilitate dynamic and adaptive interventions, improving adherence and the effectiveness of prevention strategies.

Longitudinal studies with larger and more diverse samples could explore the temporal and causal dynamics between nutritional habits, lifestyle factors, and sleep quality, overcoming the limitations of the cross-sectional analysis conducted. Moreover, the integration of objective sleep measures, such as polysomnography or tracking via wearable devices, could validate and refine the interpretation of subjective data by highlighting specific features of sleep architecture influenced by dietary and behavioral factors. Finally, the cultural and social context appears to be a key factor in the perception and interpretation of sleep quality; therefore, cross-cultural comparative studies could help to outline guidelines that are more contextualized and sensitive to the specificities of different populations. Such insights will be crucial to promote effective public health policies that integrate sleep management among the priority objectives for improving quality of life and preventing chronic diseases.

## 5. Conclusions

This study provides a significant contribution to the understanding of the relationships between nutritional habits, lifestyle factors, and sleep quality in a sample of adult Spanish women. The results highlight how specific healthy behaviors—such as a diet rich in nutrient-dense foods, regular physical activity, and a positive perception of one’s health status—are associated with better sleep quality. Conversely, alcohol consumption and nocturnal lifestyles are confirmed as risk factors for increased sleep latency and sleep fragmentation. However, some variables traditionally considered critical, such as evening consumption of stimulants or late meals, did not show significant effects, suggesting the existence of individual, genetic, or cultural adaptations that warrant further investigation. These findings underscore the importance of personalized and multidimensional strategies in promoting sleep health, with particular attention to the integrated assessment of both subjective perception and objective indicators. Future studies conducted on larger samples and using longitudinal methodologies will be essential to clarify the underlying causal mechanisms and to guide targeted interventions in clinical and public health settings.

## Figures and Tables

**Figure 1 nutrients-17-02225-f001:**
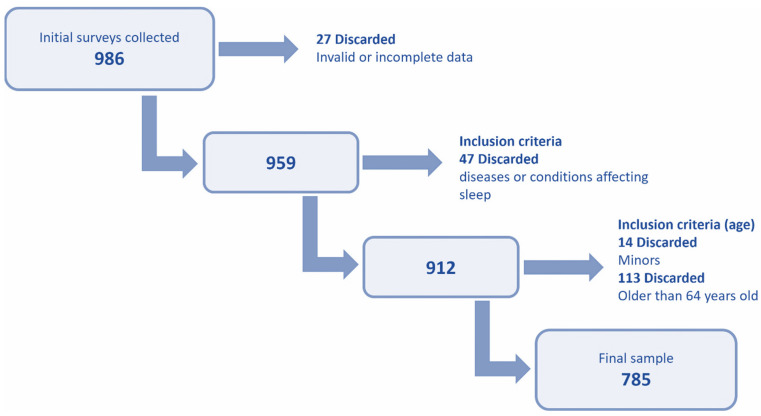
Diagram of the steps followed to obtain the final valid sample.

**Figure 2 nutrients-17-02225-f002:**
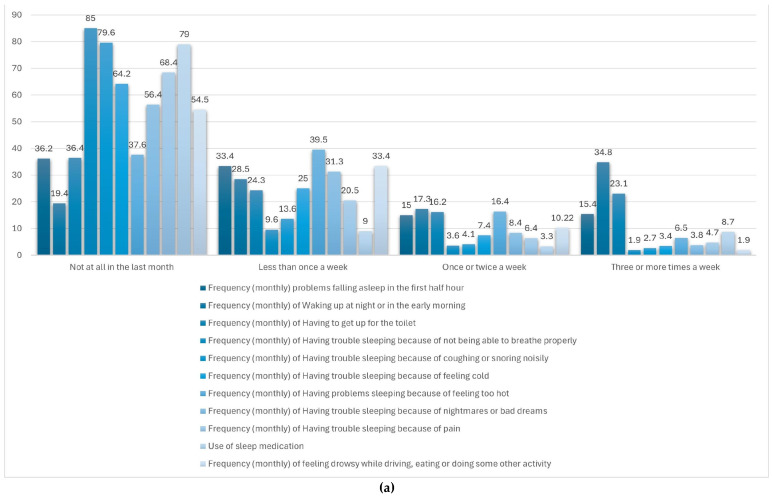
(**a**) Sleep disturbing problems. (**b**) Distribution of the components of the PSQI sleep quality scale. (**c**) Distribution of responses in the PSQI sleep quality scale.

**Figure 3 nutrients-17-02225-f003:**
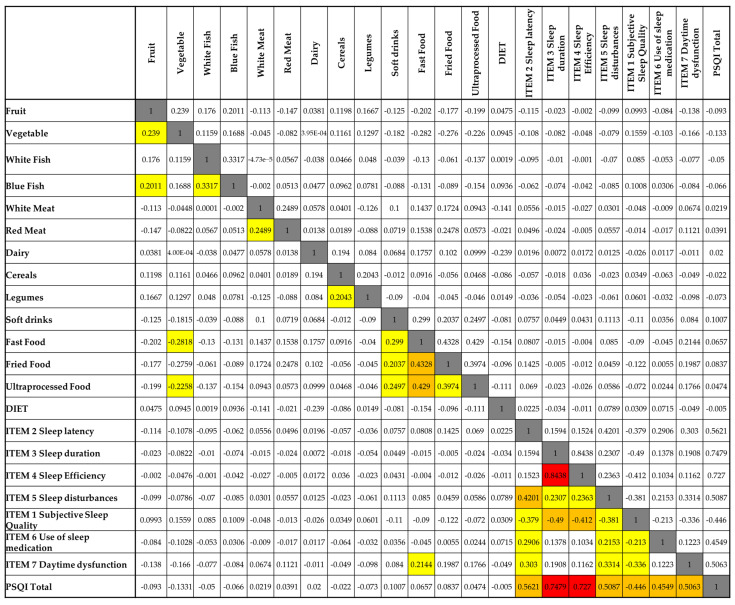
Spearman’s correlation matrix between nutritional variables and components of sleep quality. Legend: White cells = no correlation (0.00–0.19); yellow cells = low correlation (0.20–0.39); orange cells = moderate correlation (0.40–0.59); red cells = high correlation (≥0.60); grey cells = comparison of each variable with itself, which naturally results in a maximum correlation value of 1.

**Figure 4 nutrients-17-02225-f004:**
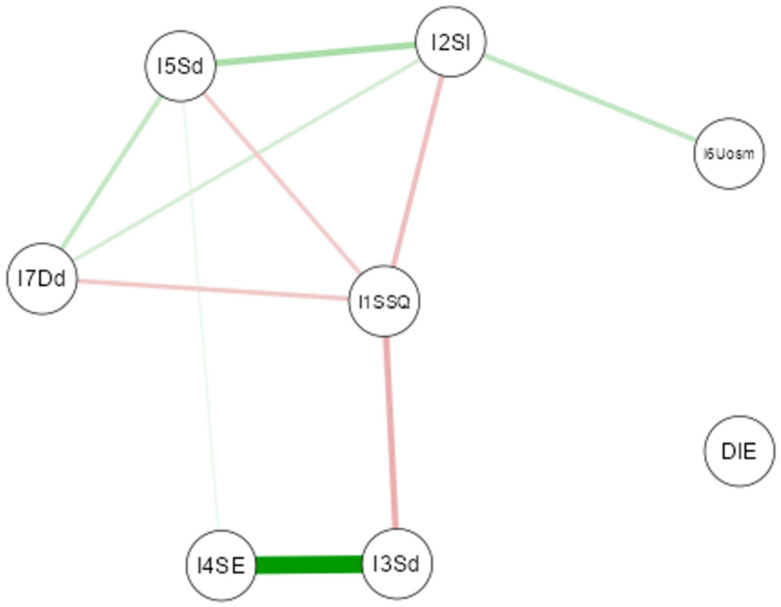
Gaussian Graphical Model of the network analysis between diet and nutritional habit variables and quality of sleep variables. (Note: The thickness of the lines reflects the magnitude of the relationship (partial correlation), while the color indicates the direction: green for positive relationships and red for negative relationships).

**Figure 5 nutrients-17-02225-f005:**
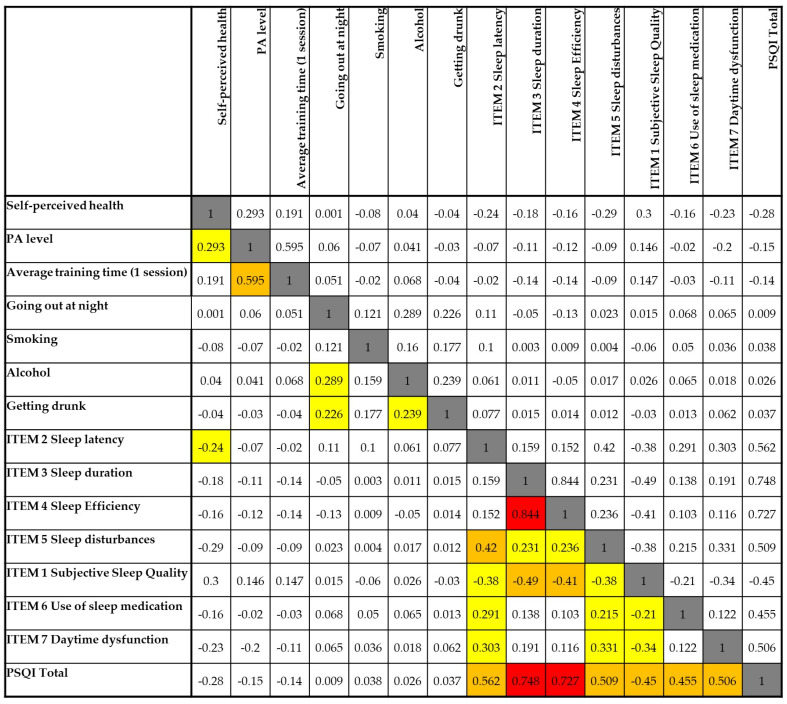
Spearman’s correlation matrix between lifestyle variables and components of sleep quality. Legend: White cells = no correlation (0.00–0.19); yellow cells = low correlation (0.20–0.39); orange cells = moderate correlation (0.40–0.59); red cells = high correlation (≥0.60); grey cells = comparison of each variable with itself, which naturally results in a maximum correlation value of 1.

**Figure 6 nutrients-17-02225-f006:**
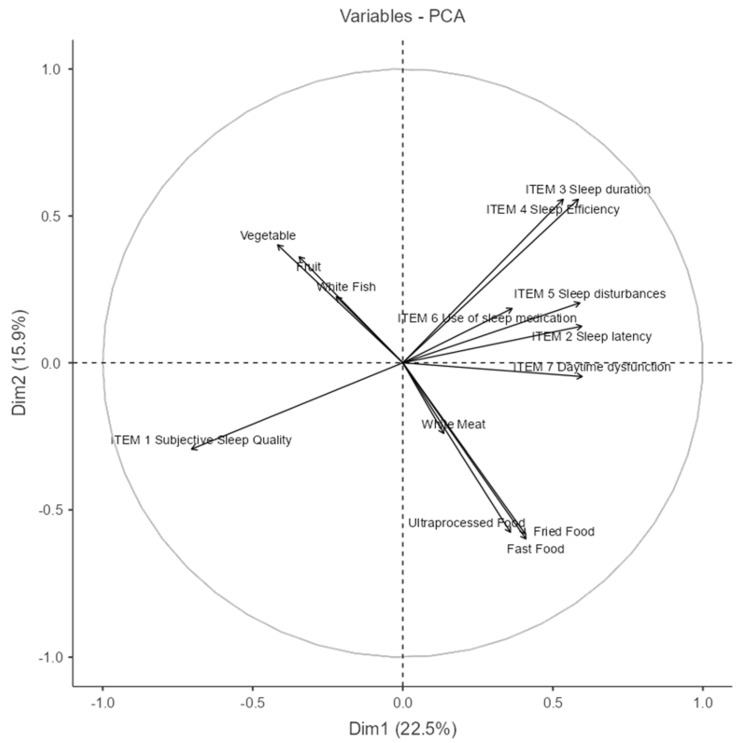
Principal Component Analysis (PCA) between nutritional and sleep quality variables. (NOTE: The circle line in a PCA correlation plot is a visual aid that shows how well each original variable is represented in the reduced 2D space and how variables relate to each other in terms of correlation structure.)

**Figure 7 nutrients-17-02225-f007:**
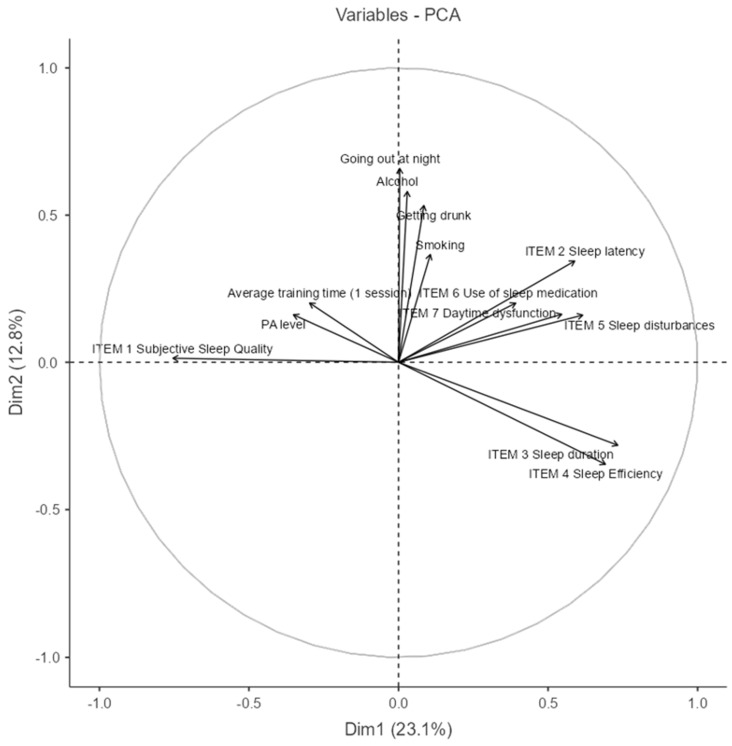
Principal Component Analysis (PCA) between lifestyle and sleep quality variables. (NOTE: The circle line in a PCA correlation plot is a visual aid that shows how well each original variable is represented in the reduced 2D space and how variables relate to each other in terms of correlation structure.)

**Table 1 nutrients-17-02225-t001:** Categorization of nutritional and lifestyle variables.

Variable	Category	Score
Fruit consumption	Never or rarely	1
	1 piece/portion per day	2
	2–4 pieces/portions per day	3
	≥5 pieces/portions per day	4
Vegetable consumption	Never or rarely	1
	1 piece/portion per week	2
	2–4 pieces/portions per week	3
	≥5 pieces/portions per week, every day	4
Sugary soft drinks	Never and very rarely (≤2 glasses per month)	4
	1 glass per week, and ≥ 2 glasses per week	3
	≤2 glasses every day	2
	3–5 glasses and >5 glasses every day	1
Fish, meat, dairy, legumes, cereals consumption	Never or very seldom	1
	1–2 times a week	2
	≥3 times a week	3
	Every day	4
Fast food, fried, ultra-processed	Never	1
	Very seldom (≤2 times a month)	2
	Once a week	3
	Several times a week	4
Getting drunk	Never or less than once a month	1
	Monthly	2
	Weekly	3
	Daily or almost daily	4
Alcohol consumption	Never or once a month	1
	2–4 times a month	2
	2–3 times a week	3
	4–5 times a week or every day	4
Smoking	Non-smoker	1
	Light smoker (<5 cigarettes per day)	2
	Moderate smoker (6–15 cigarettes per day)	3
	Severe smoker (>16 cigarettes per day)	4
Night outings	Never and sporadically	1
	1–2 times a week	2
	>3 times a week	3
	Every day	4
Time between meal and bedtime	<1 h	0
	1–2 h	1
	2–3 h	2
	>3 h	3
Dinner type	I do not usually dine	0
	Light (e.g., salad, soup, fruit)	1
	Moderate (balanced plate)	2
	Heavy (high-fat/large portions)	3
Stimulant use	No stimulant use	0
	Yes, in the morning	1
	Yes, morning and afternoon	2
	Yes, in the evening	3
Physical activity level	Very low	0
	Low (light activity <2×/week)	1
	Moderate (2–4×/week)	2
	High (≥5×/week)	3
	Very high (daily training)	4

**Table 2 nutrients-17-02225-t002:** Categorization of variables measuring sleep quality.

Variable	Category	Score
Time to go to bed	Before 10 p.m.	0
	Between 10 p.m. and 11 p.m.	1
	Between 11 p.m. and 12 a.m.	2
	Later than 12 a.m.	3
Sleep duration	>7 h	0
	6–7 h	1
	5–6 h	2
	<5 h	3
Get up time	Before 6 a.m.	0
	Between 6 a.m. and 7 a.m.	1
	Between 7 a.m. and 8 a.m.	2
	After 8 a.m.	3
Sleep efficiency	>85%	0
	75–84%	1
	65–74%	2
	<65%	3
Problems falling asleep in the first half hour (monthly frequency)	Not at all	0
	<1 a week	1
	1–2 a week	2
	≥3 times a week	3
Time needed to fall asleep (minutes)	≤15	0
	16–30	1
	31–59	2
	≥60	3
Sleep latency (frequency of problems falling asleep in the first half hour + rime needed to fall asleep)	0	0
	1–2	1
	3–4	2
	5–6	3
[Waking up at night or in the early morning; Having to get up for the toilet; Having trouble sleeping because of not being able to breathe properly; Having trouble sleeping because of coughing or snoring noisily; Having trouble sleeping because of feeling cold; Having problems sleeping because of feeling too hot; Having trouble sleeping because of nightmares or bad dreams; Having trouble sleeping because of pain] (monthly frequency)	Not at all in the last month	0
	Less than once a week	1
	Once or twice a week	2
	Three or more times a week	3
Sleep disturbances (sum of the scores obtained in the previous ITEM)	0	0
	1–9	1
	10–18	2
	19–27	3
Subjective sleep quality	Very good	0
	Fairly good	1
	Fairly poor	2
	Very poor	3
Use of sleep medication	Not at all in the last month	0
	Less than once a week	1
	Once or twice a week	2
	Three or more times a week	3
Feeling drowsy while driving, eating, or carrying out some other activity (monthly frequency)	Not at all in the last month	0
	Less than once a week	1
	Once or twice a week	2
	Three or more times a week	3
Problems in getting in the mood for any activity (monthly frequency)	No problem	0
	Only a slight problem	1
	A problem	2
	A serious problem	3
Daytime dysfunction (frequency of feeling drowsy while driving, eating or doing some other activity + frequency of problems in getting in the mood for any activity)	0	0
	1–2	1
	3–4	2
	5–6	3

**Table 3 nutrients-17-02225-t003:** Sample and their socio-demographic characteristics (N = 785).

	N	%
Age		
Young (ages 18–22)	149	19.0
Early adulthood (ages 22–34)	215	27.4
Early middle age (ages 35–44)	206	26.2
Late middle age (ages 45–64)	215	27.4
Living situation		
Alone	77	9.8
With others	708	90.2
Family living		
Without family	152	19.4
With family	633	80.6
Income		
No answer	97	12.4
Low income	255	32.5
Medium–high income	433	55.2
Education		
Basic education	299	38.1
High education	486	61.9
Healthcare education		
No	506	64.5
Yes	279	35.5

**Table 4 nutrients-17-02225-t004:** Sleep variable of participants.

**Time to go to bed**	**N**	**%**
Before 10 p.m.	60	7.60%
Between 10 p.m. and 11 p.m.	230	29.30%
Between 11 p.m. and 12 a.m.	335	42.70%
Later than 12 a.m.	160	20.40%
**Get-up time**	**N**	**%**
Before 6 a.m.	59	7.50%
Between 6 a.m. and 7 a.m.	312	39.70%
Between 7 a.m. and 8 a.m.	294	37.50%
After 8 a.m.	120	15.30%
**Time needed to fall asleep (minutes)**	**N**	**%**
≤15	416	53.00%
16–30	252	32.10%
31–59	96	12.20%
≥60	21	2.70%
**Subjective sleep quality**	**N**	**%**
Very poor	38	4.80%
Fairly poor	265	33.80%
Fairly good	430	54.80%
Very good	52	6.60%
**Frequency (monthly) of problems in getting in the mood for any activity**	**N**	**%**
No problem	428	54.50%
Only a slight problem	262	33.40%
A problem	80	10.20%
A serious problem	15	1.90%
**PSQI scale scores**	**Mean**	**SD**
Item 1: subjective sleep quality	1.63	0.68
Item 2: sleep latency	1.08	0.95
Item 3: sleep duration	1.13	0.99
Item 4: habitual sleep efficiency	0.97	1.10
Item 5: sleep disturbances	1.23	0.50
Item 6: use of sleep medication	0.42	0.91
Item 7: daytime dysfunction	1.04	0.89
**PSQI TOTAL**	**7.5**	**3.01**

**Table 5 nutrients-17-02225-t005:** Comparison of dietary habits, health indicators, and lifestyle behaviors across three groups classified by sleep quality.

	Total Sample		Good	Needs Improvement	Bad	
	Mean	(SD)	Mean (SD)	Mean (SD)	Mean (SD)	*p*-Value ^$^
Fruit	2.34	(0.77)	2.37 (0.73)	2.31 (0.81)	1.95 (0.85)	0.077
Vegetable	3.51	(0.72)	3.57 (0.67)	3.47 (0.76)	3.05 (0.91)	0.004 *
White fish	1.73	(0.57)	1.75 (0.56)	1.72 (0.58)	1.42 (0.51)	0.041 *
Blue fish	1.85	(0.57)	1.87 (0.54)	1.83 (0.59)	1.74 (0.65)	0.389
White meat	2.55	(0.67)	2.56 (0.68)	2.55(0.66)	2.32 (0.58)	0.266
Red meat	1.67	(0.67)	1.64 (0.66)	1.71 (0.67)	1.58 (0.77)	0.245
Dairy	3.44	(0.91)	3.44 (0.92)	3.43 (0.91)	3.63 (0.83)	0.522
Cereals	2.87	(0.99)	2.88 (0.98)	2.86 (1.00)	2.84 (1.07)	0.967
Legumes	2.00	(0.60)	2.03 (0.62)	1.97 (0.55)	1.84 (0.83)	0.267
Soft drinks	1.35	(0.61)	1.30 (0.55)	1.42 (0.68)	1.32 (0.48)	0.063
Fast food	2.39	(0.76)	2.37 (0.74)	2.42 (0.80)	2.47 (0.61)	0.703
Fried food	2.26	(0.82)	2.23 (0.79)	2.30 (0.84)	2.26 (0.87)	0.639
Ultra-processed food	2.48	(0.99)	2.47 (0.98)	2.46 (1.01)	2.89 (1.05)	0.184
Self-perceived health	3.79	(0.79)	3.95 (0.71)	3.63 (0.83)	3.05 (0.78)	<0.001 *
PA level	1.72	(0.87)	1.81 (0.89)	1.62 (0.82)	1.26 (0.93)	<0.001 *
Average training time	50.9	(27.8)	54.6 (28.2)	46.0 (26.3)	52.6 (32.3)	<0.001 *
Going out at night	1.17	(0.39)	1.18 (0.40)	1.16 (0.38)	1.21 (0.42)	0.654
Smoking	1.17	(0.53)	1.15 (0.49)	1.19 (0.58)	1.26 (0.56)	0.307
Alcohol	1.54	(0.74)	1.51 (0.71)	1.57 (0.79)	1.53 (0.61)	0.755
Getting drunk	1.04	(0.23)	1.03 (0.19)	1.05 (0.28)	1.11 (0.32)	<0.001 *

$ = Kruskal–Wallis test. * Statistically significant differences (*p* < 0.05).

## Data Availability

The data presented in this study are available upon reasonable request to the corresponding author. The data are not publicly available due to privacy reasons.

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
