# Peer review of "Nutritional and Lifestyle Behaviors and Their Influence on Sleep Quality Among Spanish Adult Women"

_nutrients, 2025, doi:10.3390/nu17132225_

Round 1

Reviewer 1 Report

Comments and Suggestions for Authors

Firstly, I would praise the authors since the tables and graphs are nicely done.  Although, the first table (Table 1. Categorization of nutritional and lifestyle variables) can be resized and edited to be a bit smaller. Table 2. Categorisation of variables measuring sleep quality and Table 4. Sleep variable of participants; face the same issue and should be reedited. 

The authors state that "Individuals 106 with chronic or acute medical conditions known to affect sleep quality were excluded, as 107" and later on the write that "The use of sleep medication was 280
low, with 85% of respondents indicating no usage in the past month." These statements need to be rewritten, since the usage of medication would imply that the person is suffering from a sleeping disorder. If the authors meant that other medical conditions, not sleeping disorders, were excluded, then they should give an example of which conditions were implied. 

In "Figure 2. b. Distribution of the components of the PSQI sleep quality scale." the text is overlapping itself. 

On the other hand, in the section titled "2. Materials and Methods" the total number of participants is not described, this needs to be added.

The image of figure 4. is poor quality and is unreadable. 

However, for such a long paper, there is a lack of references.

Author Response

Manuscript ID nutrients-3735126 entitled "Nutritional and Lifestyle Behaviors and Their Influence on Sleep Quality Among Spanish Adult Women”

REVIEWER  #1

N.

Editor Comment

Author Response to Comment

Page/line number

1

Firstly, I would praise the authors since the tables and graphs are nicely done.  Although, the first table (Table 1. Categorization of nutritional and lifestyle variables) can be resized and edited to be a bit smaller.

Table 2. Categorisation of variables measuring sleep quality and Table 4. Sleep variable of participants; face the same issue and should be reedited. 

Thank you very much for your positive feedback and helpful suggestions. We have carefully revised the tables as recommended. Specifically, Table 1, Table 2, and Table 4 have been resized and reformatted to be more compact and visually accessible, while maintaining clarity and readability.

Tables

2

The authors state that "Individuals 106 with chronic or acute medical conditions known to affect sleep quality were excluded, as 107" and later on the write that "The use of sleep medication was 280
low, with 85% of respondents indicating no usage in the past month." These statements need to be rewritten, since the usage of medication would imply that the person is suffering from a sleeping disorder. If the authors meant that other medical conditions, not sleeping disorders, were excluded, then they should give an example of which conditions were implied. 

Thank you for your observation. We would like to clarify that individuals with diagnosed chronic or acute sleep disorders were excluded from the study, as stated in the manuscript. The question regarding the use of sleep medication was included in the PSQI as part of the standard assessment of sleep quality, and it refers to occasional use of medication among otherwise healthy individuals. This may include those experiencing temporary periods of stress, anxiety, or other transient circumstances affecting sleep, but not individuals with a diagnosed sleep disorder. We have clarified this point in the revised version of the manuscript to avoid any misunderstanding.

“A cross-sectional study was conducted on the Spanish women population residing in Spain. Women aged between 18 and 64 years were included in the sample. Individuals with chronic or acute medical conditions known to affect sleep quality were excluded, as well as those experiencing temporary situations that could interfere with their usual routines, such as hospitalisation or incarceration. However, healthy individuals who might occasionally take some form of medication (natural or chemical) to help fall asleep were not excluded from the study.”

3

In "Figure 2. b. Distribution of the components of the PSQI sleep quality scale." the text is overlapping itself.

Thank you for your observation. We have revised Figure 2b as suggested, correcting the text overlap to ensure the labels are now clearly legible. The updated figure has been uploaded with the revised version of the manuscript.

Figure 2.b.

4

On the other hand, in the section titled "2. Materials and Methods" the total number of participants is not described, this needs to be added.

Thank you for your helpful suggestion. We have now added the total number of participants in the “2.2. Sample Size Calculation” section to provide greater clarity and transparency regarding the study sample.

2.2. Sample Size Calculation

To estimate the required sample size, reference was made to the Continuous Population Statistics (ECP) provided by the Spanish National Institute of Statistics (INE) [28]. As of 1 January 2024, the total population in Spain was 21.544.520. Based on this figure, and using the standard formula for sample size estimation in finite populations—with a 95% confidence level, 5% margin of error, and the assumption of maximum variability (p = q = 0.5)—a minimum of 385 participants was calculated to be necessary.

Finally, the total number of valid questionnaires collected was 785.

Materials and Methods

5

The image of figure 4. is poor quality and is unreadable. 

Thank you for your observation. We have replaced Figure 4 with a higher-quality version to ensure it is clear and fully readable.

Figure 4

6

However, for such a long paper, there is a lack of references.

Thank you for your observation. We have revised the manuscript, particularly the Introduction and Discussion sections, and have added appropriate and up-to-date references to support key statements that were previously unreferenced.

References

Reviewer 2 Report

Comments and Suggestions for Authors

Dear Authors,

1.Please provide a more detailed justification for the selection of the study group—women from the Spanish population.

2.Is the sample size (N=785) sufficient to draw generalizations for the population of women in Spain, and is it possible to generalize the findings to other nationalities?

3.How were potential confounding variables (e.g., age, BMI, socioeconomic status) addressed?

4.Please present the strengths and limitations of the study.

5.Please specify exactly how hormonal and/or psychological variables were controlled.

6. Please improve the presentation of Table 2 (it is too large) and Figure 3 (the font is too small).

Best regards

Reviewer

Author Response

Manuscript ID nutrients-3735126 entitled "Nutritional and Lifestyle Behaviors and Their Influence on Sleep Quality Among Spanish Adult Women”

REVIEWER  #2

N.

Editor Comment

Author Response to Comment

Page/line number

1

Please provide a more detailed justification for the selection of the study group—women from the Spanish population.

We thank the reviewer for this observation. In response, we have added the following paragraph to the Materials and Methods section to provide a more detailed justification for the selection of the study group:

“The study focused on adult women because, on the one hand, women tend to show greater willingness and reliability when responding to health and nutrition-related questionnaires. Previous research has consistently shown that women are more engaged in health-related studies and more likely to report their behaviors and symptoms accurately, which contributes to more robust and interpretable data [23-25]. On the other hand, scientific evidence from the Spanish adult population indicates that women experience significantly higher rates of sleep disorders, such as insomnia, and poorer sleep quality compared to men [26,27]. Therefore, targeting this population makes the study particularly relevant.”

Materials and Methods

2

Is the sample size (N=785) sufficient to draw generalizations for the population of women in Spain, and is it possible to generalize the findings to other nationalities?

We thank the reviewer for this observation. We believe that the sample size (N = 785) is sufficiently large to allow generalizations to be made for the adult female population in Spain, as it more than doubles the minimum required sample size calculated for this population with a 95% confidence level and 5% margin of error.  We have added the paragraph below to the Materials and Methods section to provide a more detailed justification for the sample size calculation.

Regarding the possibility of extending the findings to other nationalities, we consider that the results may be relevant and potentially generalizable to adult women in other Western countries that share similar socioeconomic conditions and lifestyle patterns with Spain. However, we acknowledge that cultural, environmental, and societal differences between countries could influence sleep behaviors and quality. Therefore, we agree that to assess the specific impact of nationality or country of residence on sleep quality, dedicated cross-national or comparative studies would be necessary.

2.2. Sample Size Calculation

To estimate the required sample size, reference was made to the Continuous Population Statistics (ECP) provided by the Spanish National Institute of Statistics (INE) [D]. As of 1 January 2024, the total population in Spain was 21.544.520. Based on this figure, and using the standard formula for sample size estimation in finite populations—with a 95% confidence level, 5% margin of error, and the assumption of maximum variability (p = q = 0.5)—a minimum of 385 participants was calculated to be necessary.

Sample Size Calculation

3

How were potential confounding variables (e.g., age, BMI, socioeconomic status) addressed?

Thank you for your valuable comment. In order to explore the potential influence of confounding variables such as age, BMI, or socioeconomic status, we attempted a cluster analysis to identify whether the sample could be segmented into distinct groups based on these variables. However, the analysis did not yield any meaningful or interpretable clusters, suggesting that there were no clearly separable patterns within the sample according to these variables.

Given that the primary aim of the study was exploratory and descriptive, we did not proceed with multivariate adjustments. Nonetheless, we acknowledge this as a limitation and have added a note in the “Limitations” section accordingly.

“Additionally, given the exploratory and descriptive nature of the study, multivariate adjustments were not performed. This may limit the ability to control confounding effects and should be considered when interpreting the results.”

Strengths and limitations

4

Please present the strengths and limitations of the study.

Thank you for the suggestion. We have added a concise paragraph at the end of the Discussion outlining the main strengths and limitations of the study.

Strengths and limitations

5

Please specify exactly how hormonal and/or psychological variables were controlled.

We appreciate the reviewer’s insightful comment. In our study, no specific control was applied for hormonal or psychological variables, as the questionnaire employed did not include items exploring these dimensions. This choice was deliberate, as the authors aimed to focus primarily on dietary behaviors and patterns, without delving into psychological aspects that fall outside the scope of our expertise.

Nevertheless, we would like to clarify that individuals with clinically diagnosed eating disorders or severe medical conditions were excluded from participation at the outset of the study. This criterion was established to reduce potential confounding effects related to such conditions.

In light of the reviewer’s comment, we have added a sentence to the Limitations section to explicitly acknowledge the lack of analysis on hormonal variables, and to suggest this as an avenue for future research.

“Moreover, the study did not include hormonal and psychological variables, which are known to influence sleep in women and could have provided additional insight into the mechanisms underlying sleep disturbances in this population. Future studies should consider integrating these dimensions into the analysis.”

Strengths and limitations

6

Please improve the presentation of Table 2 (it is too large) and Figure 3 (the font is too small).

Thank you for your valuable feedback. Following your suggestion, and in line with the comments of another reviewer, we have revised several tables and figures to enhance their clarity and legibility.

Table 2 and Figure 3